# Chronic Gut Inflammation and Dysbiosis in IBS: Unraveling Their Contribution to Atopic Dermatitis Progression

**DOI:** 10.3390/ijms25052753

**Published:** 2024-02-27

**Authors:** Jae-Hwan Jang, Sun-Young Jang, Sora Ahn, Ju-Young Oh, Mijung Yeom, Seok-Jae Ko, Jae-Woo Park, Soon-Kyeong Kwon, Kyuseok Kim, In-Seon Lee, Dae-Hyun Hahm, Hi-Joon Park

**Affiliations:** 1Jaseng Spine and Joint Research Institute, Jaseng Medical Foundation, Seoul 05854, Republic of Korea; jjh2392@jaseng.org; 2Department of Science in Korean Medicine, Graduate School, Kyung Hee University, Seoul 02447, Republic of Korea; jsy9885@khu.ac.kr; 3Department of Anatomy and Information Sciences, College of Korean Medicine, Kyung Hee University, Seoul 02447, Republic of Korea; lljasmintll@gmail.com (S.A.); ohjuyoung@khu.ac.kr (J.-Y.O.); 4Acupuncture & Meridian Science Research Center, Kyung Hee University, Seoul 02447, Republic of Korea; myeom@khu.ac.kr (M.Y.); inseon.lee@khu.ac.kr (I.-S.L.); dhhahm@khu.ac.kr (D.-H.H.); 5Department of Gastroenterology, College of Korean Medicine, Kyung Hee University, Seoul 02447, Republic of Korea; kokokoko119@hanmail.net (S.-J.K.); pjw2907@hanmail.net (J.-W.P.); 6Division of Applied Life Science (BK21), Gyeongsang National University, Jinju 52828, Republic of Korea; skkwon@gnu.ac.kr; 7Department of Ophthalmology, Otorhinolaryngology, and Dermatology of Korean Medicine, College of Korean Medicine, Kyung Hee University, Seoul 02447, Republic of Korea; kuseok@hanmail.net; 8Department of Meridian & Acupoint, College of Korean Medicine, Kyung Hee University, Seoul 02447, Republic of Korea; 9Department of Physiology, School of Medicine, Kyung Hee University, Seoul 02447, Republic of Korea; 10Department of KHU-KIST Convergence Science & Technology, Kyung Hee University, 26 Kyungheedae-ro, Dongdaemun-gu, Seoul 02447, Republic of Korea

**Keywords:** atopic dermatitis, gut microbial dysbiosis, systemic inflammation, irritable bowel syndrome

## Abstract

Emerging evidence suggests a link between atopic dermatitis (AD) and gastrointestinal disorders, particularly in relation to gut microbial dysbiosis. This study explored the potential exacerbation of AD by gut inflammation and microbial imbalances using an irritable bowel syndrome (IBS) mouse model. Chronic gut inflammation was induced in the model by intrarectal injection of 2,4,6-trinitrobenzene sulfonic acid (TNBS), followed by a 4-week development period. We noted significant upregulation of proinflammatory cytokines in the colon and evident gut microbial dysbiosis in the IBS mice. Additionally, these mice exhibited impaired gut barrier function, increased permeability, and elevated systemic inflammation markers such as IL-6 and LPS. A subsequent MC903 challenge on the right cheek lasting for 7 days revealed more severe AD symptoms in IBS mice compared to controls. Further, fecal microbial transplantation (FMT) from IBS mice resulted in aggravated AD symptoms, a result similarly observed with FMT from an IBS patient. Notably, an increased abundance of *Alistipes* in the feces of IBS mice correlated with heightened systemic and localized inflammation in both the gut and skin. These findings collectively indicate that chronic gut inflammation and microbial dysbiosis in IBS are critical factors exacerbating AD, highlighting the integral relationship between gut and skin health.

## 1. Introduction

Atopic dermatitis (AD), a chronic inflammatory skin disease, has been increasingly associated with systemic health conditions, particularly gastrointestinal (GI) disorders [1]. It is primarily characterized by skin erythema, edema, and dryness, driven by an increased T-helper (Th) 2 cell-mediated immune response [1]. The etiology of AD involves a complex interplay of genetic, environmental, epidermal barrier dysfunction, and abnormal immune responses [2]. Furthermore, recent epidemiological studies have elucidated the prevalence and impact of AD globally, emphasizing its rising incidence and the burden it places on individuals and healthcare systems [3].

Emerging research indicates that the etiology of AD, while multifactorial, is strongly influenced by epidermal barrier dysfunction and abnormal immune responses. Intriguingly, emerging research underscores the significant role of gut health, particularly the gut microbiota balance, in the pathogenesis and exacerbation of AD [4]. Changes in gut microbiota associated with AD involve a reduction in beneficial bacteria and an increase in pathogenic microbes, potentially contributing to systemic inflammation and skin barrier dysfunction in AD patients. This connection opens new avenues for understanding and potentially treating AD, emphasizing the importance of the gut–skin axis [5].

The gut microbiome, a complex ecosystem of microorganisms, is known to influence systemic immunity and inflammation beyond the gastrointestinal tract [6]. Gut microbial dysbiosis, an imbalance in this microbial community, has been linked to various inflammatory conditions, including those affecting skin diseases such as AD [7]. A notable example is irritable bowel syndrome (IBS), characterized by persistent gut inflammation and changes in gut microbiota [8]. Research indicates that in conditions such as IBS, gut inflammation can lead to increased intestinal permeability. This change allows for microbial products and pro-inflammatory cytokines to enter the systemic circulation, possibly contributing to widespread inflammation [9]. This mechanism can exacerbate conditions like AD. The higher prevalence of AD in patients with gut inflammatory diseases, including IBS, underscores the importance of the gut–skin axis [10,11,12]. However, the specific mechanisms and the extent to which gut inflammation and microbial dysbiosis influence AD remain underexplored.

In this study, we investigate the potential exacerbation of AD by gut inflammation and microbial imbalances using an IBS mouse model. Through this model, we aim to elucidate the interactions between gut health and skin inflammation, specifically examining how chronic gut inflammation and microbial dysbiosis in IBS can influence the severity of AD. Our findings seek to provide deeper insights into the gut–skin axis, potentially paving the way for novel therapeutic strategies in managing AD.

## 2. Results

### 2.1. GI Dysfunction in the TNBS-Induced IBS Model

To establish the IBS model, we administered 2,4,6-trinitrobenzene sulfonic acid (TNBS) intrarectal (i.r.) injections to mice and allowed a 4-week period for the development of the IBS model. The CON group was injected with ethanol (i.r.). The detailed schedule is shown in Figure 1A. Body weight was measured on Days 7 and 28 (two-way repeated-measures analysis of variance (ANOVA), Time, F_2,52_ = 151.4, *p* < 0.0001; Group, F_1,53_ = 18.37, *p* = 0.0006). Results showed that the mean body weight of the TNBS-induced IBS mice was significantly lower compared to that of the CON group mice at Day 28 (Bonferroni’s post hoc test, *p* < 0.001; Figure 1B). Next, GI dysfunction was evaluated by visualizing the stool morphology and bead expulsion test. Following TNBS administration, diarrhea was induced on Day 3; however, on Day 28, the stool morphology changed, and diarrhea was replaced with constipation (Figure 1C). Moreover, the bead expulsion latency was lower in the IBS group compared to that in the CON group (unpaired two-tailed *t*-test, t_22_ = 3.594, *p* < 0.001; Figure 1D). To investigate the inflammatory status in our mice model [13], we performed hematoxylin and eosin (H&E) staining in samples obtained from the distal colon. Results showed that the length of crypts was shorter in the IBS group compared to that in the CON group (unpaired two-tailed *t*-test, t_8_ = 4.083, *p* = 0.0035), while the inflammation score was higher (unpaired two-tailed *t*-test, t_8_ = 7.750, *p* < 0.0001; Figure 1E–G). In addition, the levels of inflammatory markers such as *interleukin (Il)-6* (unpaired two-tailed *t*-test, t_5_ = 6.335, *p* = 0.0007), *Mmp12* (unpaired two-tailed *t*-test, t_6_ = 5.155, *p* = 0.0021), *Il-31* (unpaired two-tailed *t*-test, t_6_ = 5.369, *p* = 0.0017), *Il-22* (unpaired two-tailed *t*-test, t_6_ = 5.437, *p* = 0.0016), and *Ifn-r* (unpaired two-tailed *t*-test, t_6_ = 4.086, *p* = 0.0065) were also elevated in the colon of the IBS group compared to the CON group (Figure 1H–L).

### 2.2. Leaky Gut and Alterations in the Skin of the IBS Model

To investigate the importance of the gut–skin axis, we assessed leaky gut and systemic inflammation in our mice model. Increased gut permeability and loss of gut barrier integrity cause a release of several pathogenic substances into the blood, including endotoxins and pro-inflammatory cytokines [14]. This induces systemic inflammation that may lead to several adverse effects by delivering the inflammatory molecules to various organs in the body. The expression levels of tight junction proteins in the distal colon were measured to confirm the loss of the gut barrier function in the TNBS-induced IBS mice. Zonula occludens (ZO-1) (unpaired two-tailed *t*-test, t_8_ = 4.822, *p* = 0.0013) and occludin (unpaired two-tailed *t*-test, t_8_ = 4.544, *p* = 0.0019) expression levels were lower in the IBS group compared to those in the CON group (Figure 1M–O). Next, we measured the blood level of dextran-FITC 4 h after oral administration and found that the IBS group showed a 2.3-fold increase in the dextran level compared to the CON group, indicative of the increase in gut permeability (unpaired two-tailed *t*-test, t_16_ = 8.255, *p* < 0.0001; Figure 1P). Moreover, the serum levels of IL-6 (unpaired two-tailed *t*-test, t_14_ = 4.311, *p* = 0.0007) and lipopolysaccharide (LPS) (unpaired two-tailed *t*-test, t_8_ = 4.085, *p* = 0.0035) were significantly higher in the IBS group than in the CON group, whereas the expression level of tumor necrosis factor-α (TNF-α) showed no difference between the two groups (unpaired two-tailed *t*-test, t_14_ = 0.4516, *p* = 0.6585; Figure 1Q–S).

Finally, to investigate the effect of gut inflammation on the skin, we measured the levels of inflammatory cytokines in the mouse skin. Thymic stromal lymphopoietin (TSLP) has been known to be highly expressed in the lesioned skin of patients with AD and is a representative inflammatory cytokine present in the skin [15]. The TSLP level (unpaired two-tailed *t*-test, t_8_ = 4.061, *p* = 0.0036) in the skin was significantly higher in the IBS group compared to that in the CON group (Figure 1T,U). IL-6 binds to IL-6R and regulates various functions, such as immune response, and is known to play a role in repairing skin damage [16]. The expression level of IL-6 receptor (IL-6R) (unpaired two-tailed *t*-test, t_8_ = 7.122, *p* < 0.0001) was increased in the mice skin in IBS mice (Figure 1T,V).

### 2.3. Exacerbation of AD Symptoms by IBS

To determine whether increased gut permeability aggravated the AD symptoms in the skin, mice were exposed to MC903 for 7 days and AD-like symptoms were evaluated in the IBS model (Figure 2A). The severity of skin lesions was evaluated using the modified scoring atopic dermatitis (SCORAD) score, which is used to evaluate the severity of atopic skin lesions in clinical practice [17] (Figure 2B,C). The modified SCORAD score was higher in the IBS+AD group than in the AD group (unpaired two-tailed *t*-test, t_13_ = 4.159, *p* = 0.0011; Figure 2C). The duration of scratching on both Day 5 (one-way ANOVA, F_3,28_ = 8.920, *p* = 0.0003) and Day 10 (one-way ANOVA, F_3,28_ = 5.020, *p* = 0.0066) was higher in the IBS+AD group compared to that in the AD group (Newman–Keuls post hoc test, Day 5, *p* < 0.05; day 10, *p* < 0.05; Figure 2D,E). The epidermal thickness, a representative histological feature of AD, was also increased in the IBS+AD group compared to that in the AD group (one-way ANOVA, F_3,16_ = 53.39, *p* < 0.0001; Newman–Keuls post hoc test, *p* < 0.001; Figure 2F,G). Moreover, the expression level of TSLP in the epidermis was higher in the IBS+AD group than in the AD group (one-way ANOVA, F_3,16_ = 63.73, *p* < 0.0001; Newman–Keuls post hoc test, *p* < 0.001) and the CON group (*p* < 0.001; Figure 2F,H).

### 2.4. Exacerbation of AD Symptoms in IBS Mice–Fecal Microbiota-Transplanted Mice

To investigate whether gut microbial dysbiosis deteriorates the development of AD, we performed fecal microbial transplantation (FMT) from CON and IBS mice to antibiotic-treated mice (Figure 3A). Then, we investigated GI dysfunction through the size of fecal pellets and the bead expulsion test in mice transplanted with fecal microbiota from the CON group (C-MF) and mice transplanted with fecal microbiota from the IBS group (I-MF). The size of fecal pellets was smaller, and the bead expulsion latency was higher in the I-MF group than in the C-MF group (unpaired two-tailed *t*-test, t_26_ = 2.174, *p* = 0.0388; Figure 3B,C). Moreover, H&E staining was performed to examine the inflammatory status in the distal colon. Results showed that the crypt lengths were lower in the I-MF group than that in the C-MF group (unpaired two-tailed *t*-test, t_8_ = 2.881, *p* = 0.0205; Figure 3D,E), while the inflammation score was higher (unpaired two-tailed *t*-test, t_8_ = 7.419, *p* < 0.0001; Figure 3D,F). To investigate the effect of gut microbial dysbiosis on gut barrier functions, we examined the expression levels of tight junction proteins such as ZO-1 and occludin in the colon. Results showed that the levels of ZO-1 (unpaired two-tailed *t*-test, t_8_ = 3.457, *p* = 0.0086) and occludin (unpaired two-tailed *t*-test, t_8_ = 3.923, *p* = 0.0044) were lower in the I-MF group than those in the C-MF group (Figure 3G–I). Furthermore, serum levels of IL-6 (unpaired two-tailed *t*-test, t_8_ = 5.704, *p* = 0.0005), TNF-α (unpaired two-tailed *t*-test, t_8_ = 2.747, *p* = 0.0252), and LPS (unpaired two-tailed *t*-test, t_8_ = 4.969, *p* = 0.0011) were significantly higher in the I-MF group than in the C-MF group (Figure 3J–L). In addition, skin Il-6R levels (unpaired two-tailed *t*-test, t_8_ =2.694, *p* = 0.0273) were also significantly higher in the I-MF group (*p* < 0.001; Figure 3M,N).

Next, we investigated whether FMT-induced gut microbial dysbiosis contributes to the exacerbation of AD symptoms. The modified SCORAD score was higher in the AD+I-MF group than that in the AD+C-MF group (unpaired two-tailed *t*-test, t_15_ = 5.418, *p* < 0.0001; Figure 3O,P). The total duration of scratching on both Day 5 (one-way ANOVA, F_3,28_ = 37.10, *p* < 0.0001) and Day 10 (one-way ANOVA, F_3,28_ = 64.72, *p* < 0.0001) was higher in the AD+I-MF group than in the CON+C-MF group (Newman–Keuls post hoc test, Day 5, *p* < 0.001; day 10, *p* < 0.001). Interestingly, the total duration of scratching was significantly longer in the AD+I-MF group than in the AD+C-MF group (Day 5, *p* < 0.01; day 10, *p* < 0.01; Figure 3Q,R). Epidermal thickness was increased in the AD+I-MF group compared to that in AD+C-MF (one-way ANOVA, F_3,16_ = 107.7, *p* < 0.0001; Newman–Keuls post hoc test, *p* < 0.001; Figure 3S,T). Moreover, the expression level of TSLP in the epidermis was higher in the AD+I-MF group than that in AD+C-MF (one-way ANOVA, F_3,16_ = 20.68, *p* < 0.0001; Newman–Keuls post hoc test, *p* < 0.001; Figure 3S,U). These results confirmed the exacerbation of AD symptoms in mice following transplantation of gut microbiota from animals with gut microbial dysbiosis.

### 2.5. Gut Microbial Dysbiosis and Its Association with Gut–Blood–Skin Axis Markers in IBS Mice

Leaky gut and gut microbial dysbiosis are associated with GI disorders such as IBS and inflammatory bowel disease (IBD), and a high incidence of AD has been reported in patients with these diseases [11,18,19]. To investigate gut microbial dysbiosis in the TNBS-induced IBS model, we performed 16s rRNA gene sequencing in mouse fecal samples. An average of 24,286 reads were obtained per sample, and 905 discrete bacterial taxa (operational taxonomic units, OTUs) were identified. The α-diversity estimating microbial community richness did not show a significant difference between CON and IBS groups based on the observed OTUs (CON, 275.0 ± 17.0; IBS, 320.2 ± 23.6, *p* = 0.222), the Shannon index (CON, 4.1 ± 0.1; IBS, 4.3 ± 0.1, *p* = 0.136), and the Inverse Simpson index (CON, 27.5 ± 4.8; IBS, 38.3 ± 4.9, *p* = 0.07702). The extent of similarity in gut microbial communities between CON and IBS groups was measured using the principal coordinate analysis (PCoA) based on Bray–Curtis (ANOSIM R = 0.0996, *p* = 0.045), UniFrac unweighted (ANOSIM R = 0.0662, *p* = 0.193) and UniFrac weighted (ANOSIM R = 0.0941, *p* = 0.151) distances at the OTU level. In Bray–Curtis cluster analysis, the gut microbiota of the groups was distinctly separated, suggesting that the overall structure of the bacterial community in the CON and IBS groups was significantly different (Figure 4A). A significant shift in the microbiota based on the relative abundance is shown in the cladogram (Figure 4B). Furthermore, these LEfSe comparisons identified 21 taxa that were differentially abundant between the two groups (Figure 4C). At the genus level, the relative abundance of *Alistipes* was significantly increased in the IBS group (unpaired two-tailed *t*-test, t_8_ = 9.678, *p* < 0.001; Figure 4D). The heat map shows a correlation between gut–blood–skin axis markers. Among them, *Alistipes* showed a high correlation with gut–blood–skin axis markers (vs. colon ZO-1, r = −0.8080, *p* = 0.0047; vs. colon Occludin, r = −0.8977, *p* = 0.0004; vs. serum IL-6, r = 0.6371, *p* = 0.0476; vs. serum TNF-α, r = 0.1306, *p* = 0.7192; vs. serum LPS, r = 0.7533, *p* = 0.0119; vs. gut permeability, r = 0.9292, *p* = 0.0001; vs. skin TSLP, r = 0.7837, *p* = 0.0073; vs. skin IL-6R, r = 0.9294, *p* < 0.0001; Figure 4E,F).

### 2.6. Aggravation of AD Symptoms in IBS Patient–Fecal Microbiota-Transplanted Mice

To confirm whether gut microbial dysbiosis in the IBS patient affects the aggravation of AD, we transplanted fecal microbiota from the IBS patient into mice (Figure 5A). H&E staining was performed to examine inflammatory status in the distal colon of mice transplanted from the gut microbiota of healthy control (C-HF) and mice transplanted from the gut microbiota of the IBS patient (I-HF). Results showed that the length of crypts was lower in the I-HF group than in the C-HF group (unpaired two-tailed *t*-test, t_8_ = 7.251, *p* < 0.0001; Figure 5B,C), while the inflammation score was higher (unpaired two-tailed t-test, t_8_ = 6.736, *p* = 0.0001; Figure 5B,D). To investigate the loss of the gut barrier function, the expression levels of tight junction proteins in the distal colon were evaluated. Results showed that the levels of ZO-1 (unpaired two-tailed *t*-test, t_8_ = 3.457, *p* = 0.0086) and occludin (unpaired two-tailed t-test, t_8_ = 3.923, *p* = 0.0044) were lower in the I-HF group than in the C-HF group (Figure 5E–G). Moreover, serum levels of IL-6 (unpaired two-tailed *t*-test, t_8_ = 5.704, *p* = 0.0005) and LPS (unpaired two-tailed *t*-test, t_8_ = 4.969, *p* = 0.0011) were significantly higher in the I-HF group than in the C-HF group (Figure 5H–J). In addition, our results showed that the skin levels of il-6r mRNA (unpaired two-tailed *t*-test, t_6_ = 3.305, *p* = 0.0163) were higher in the I-HF group than in the C-HF group (Figure 5K). The expression levels of *Gata3* (unpaired two-tailed *t*-test, t_6_ = 3.951, *p* = 0.0075) and *Il-4r* (unpaired two-tailed *t*-test, t_6_ = 2.832, *p* = 0.0299) were higher in the I-HF group (Figure 5L,M). Also, the expression levels of IL-6R (unpaired two-tailed *t*-test, t_8_ =2.192, *p* = 0.0597) were increased in the skin of I-HF group mice (Figure 5N,O).

Next, to investigate whether AD symptoms aggravated in I-HF mice, we examined scratching behavior following MC903 application. The total durations of scratching on both Day 5 (one-way ANOVA, F_3,29_ = 23.67, *p* < 0.0001) and Day 10 (one-way ANOVA, F_3,29_ = 37.03, *p* < 0.001) were higher in the AD+I-HF group than that in the CON+C-HF group (Newman–Keuls post hoc test, Day 5, *p* < 0.001; Day 10, *p* < 0.001; Figure 5P,Q). Epidermal thickness was also increased in the AD+I-HF group compared to that in AD+C-HF (one-way ANOVA, F_3,32_ = 35.28, *p* < 0.0001; Newman–Keuls post hoc test, p < 0.001; Figure 5R,S). Moreover, the expression levels of TSLP in the epidermis were higher in the AD+I-HF group than in AD+C-HF (one-way ANOVA, F_3,32_ = 35.28, *p* < 0.0001; Newman–Keuls post hoc test, *p* < 0.001; Figure 5R,T). These results confirmed that AD symptoms are exacerbated in mice following transplantation of gut microbiota from the patient with IBS.

## 3. Discussion

Our study offers pivotal insights into the gut–skin axis, particularly in the context of AD. Using an IBS mouse model, we confirmed and expanded upon the hypothesis that gastrointestinal disorders, especially those associated with gut microbial dysbiosis, can aggravate AD. The marked upregulation of pro-inflammatory cytokines and the dysbiosis observed in the IBS model underscore a mechanistic link, enhancing our understanding of how gastrointestinal disorders could influence AD.

In this study, to explore the gut–skin relationship, we first established an IBS mouse model to simulate chronic mild gut inflammation. After a one-month development period following the TNBS challenge, we observed an increase in intestinal inflammation, a decrease in crypt length, and reduced levels of tight junction proteins (ZO-1 and occludin). In addition, not only gut permeability but the serum levels of IL-6 and LPS were also significantly elevated, suggesting leaky gut and systemic inflammation also occurred. 

Next, we explored the impact of chronic gut inflammation on skin condition in the IBS mice. We adopted an MC903-induced AD model, which has been known to induce AD-like inflammation, mimicking most of the clinical, histological, and immunological features shown in AD patients [20]. Surprisingly, we observed that the levels of TSLP and IL-6R were markedly elevated in the skin epidermis, even in the absence of an MC903 challenge. Furthermore, it was noted that AD symptoms, as evidenced by increased scratching behaviors and higher SCORAD scores, worsened in IBS mice following MC903 challenges. These findings align with systematic reviews and meta-analyses that have reported a higher prevalence of AD in patients with IBS and IBD compared to the general population [11,21]. This observation underscores the role of gut inflammation in the pathogenesis of AD and suggests that gut-to-skin effects are significant.

As a next step, we aimed to elucidate the role of gut microbiota in regulating the gut–skin axis, particularly in the context of AD development. One of the most striking findings of our study was the exacerbation of AD symptoms following FMT from the IBS-affected mice. This result not only highlights the significant role of gut microbial dysbiosis in AD progression, but also implies a direct influence of gut microbiota composition on skin health. Then, to investigate why gut microbiota of IBS mice could result in exacerbated AD symptoms, we performed 16S rRNA gene sequencing of fecal samples from IBS-afflicted mice. We discovered significant alterations in their microbial community composition compared to CON mice. Notably, there was an increased gut abundance of several bacterial groups, including *Rikenellaceae*, *Alistipes*, *Lactobacillales*, *Lactobacillaceae*, *Lactobacillus*, *Bacilli*, *Tenericutes*, *Mollicutes*, *Acholeplasma*, *Acholeplasmataceae*, *Ruthenibacterium*, and *Eubacterium* in the IBS mouse. In contrast, we observed a reduction in *Parabacteteroides*, *Porphyromonadaceae*, *Erysipelotrichaceae*, *Erysipelotrichi*, and *Erysipelotrichales*. It is noteworthy that similar increases in the gut abundance of *Rikenellaceae*, *Alistipes*, *Tenericutes*, and *Eubacterium* have been observed in human IBS patients [22,23,24,25], as well as increases in that of *Lactobacillus* and *Bacilli* in patients with IBD [26,27]. These findings align with existing research, highlighting the potential significance of these bacterial groups in GI disorders. Further analysis revealed a significant correlation between the altered gut microbiota, specifically *Alistipes*, and markers of gut integrity, gut permeability: its abundance was inversely correlated with the integrity of tight junctions in the gut epithelia and positively correlated with increased gut permeability, as well as elevated serum levels of LPS and IL-6. In addition, we observed that the altered gut microbiota can be the prognostic factors to predict the pathogenesis of AD: the abundance of *Alistipes* was positively associated with the expressions of TSLP in the skin, suggesting a close influence of gut microbiota on skin inflammation. This aligns with previous literature indicating that *Alistipes* is associated with abdominal pain [28] and may play a role in systemic inflammation and increased gut permeability [29,30,31], both of which are known to exacerbate AD features. This finding is particularly intriguing as it links a specific microbial genus with physiological changes relevant to exacerbation of AD. Nevertheless, further in-depth research is essential to elucidate the specific role that *Alistipes* plays in the aggravation of atopic dermatitis. 

To further explore the clinical significance of our findings, we transplanted fecal microbiota from an IBS patient to an AD mice and observed that AD phenotypes worsened compared to FMT from a healthy subject. The results were consistent with those observed in our animal model. This suggests that our research findings may have clinical relevance. Overall, our study presents novel insights by demonstrating that gut inflammation and gut dysbiosis shown in IBS can exacerbate AD, while previous studies have explored their association [11,12,32]. Thus, we suggest that modulating gut microbiota could be a new therapeutic approach for AD.

The exact mechanism by which gut microbial dysbiosis and inflammation influence skin condition remains unclear. However, it is hypothesized that increased serum levels of IL-6, rather than TNF-α, might play a crucial role. In our study, we observed elevated levels of IL-6 in the serum and an increased expression of IL-6R in the skin layers of IBS mice. These increases were also replicated in mice that underwent FMT from both the IBS mouse and patient. It is believed that IL-6 is produced through the activation of epithelial–macrophage crosstalk in the colon, triggered by gut inflammation and microbial dysbiosis [33,34]. This could enhance the concentration of IL-6 in the bloodstream due to an increase in gut permeability. IL-6 binds to the IL-6R and regulates various functions by activating IL-6 trans-signaling in the skin [35]. It can initiate the development of Th2 cells from Th0 cells [36], and IL-6 trans-signaling is known to promote Th2-mediated AD [37]. Activation of Th2 cells can alter cell-mediated immune responses and promote IgE-mediated hypersensitivity, which plays a crucial role in the development of AD [38]. In this study, we also observed the increased expression of Th2-related genes, *Gata3* and *Il-4r*, in the skin of FMT mice from the IBS patient, supporting the role of IL-6 and IL-6R in the development of the AD phenotype in mice.

While our study provides substantial evidence linking gut inflammation and microbial dysbiosis with AD development, several limitations remain. The precise mechanisms of how gut dysbiosis contributes to systemic inflammation and the detailed role of the gut–skin axis in AD pathogenesis need further exploration. Additionally, understanding the interaction between inflammatory factors like IL-6 and IL-6R in AD development is crucial. In addition, the extrapolation of results from an IBS mouse model to human AD patients requires cautious interpretation. While our findings offer valuable insights, the complexities of human physiology and the multifactorial nature of AD mean that these results should be considered a starting point for further research. Additionally, the exact mechanisms by which gut microbial dysbiosis and inflammation influence skin conditions in humans remain to be fully elucidated.

In conclusion, our research highlights the intricate relationship between chronic gut inflammation, microbial dysbiosis, and the exacerbation of AD. These findings provide a foundation for future investigations into the gut–skin axis and open up new possibilities for treating AD by targeting gut health. As AD remains a significant challenge in dermatology, understanding these complex interactions offers hope for more effective and comprehensive treatment approaches.

## 4. Materials and Methods

### 4.1. Animals

C57BL/6J male mice [6 weeks old; weight, 20–23 g; DBL, Eumseong, Republic of Korea] were housed individually at a temperature of 22 ± 2 °C under a 12/12 h light/dark cycle [light: 08:00 a.m. to 08:00 p.m., dark: 08:00 p.m. to 08:00 a.m.] with free access to food and water for at least seven days before the commencement of experiments. All experimental protocols were approved by the Kyung Hee University Animal Care Committee for Animal Welfare [KHSASP-22-015]. Detailed information on group allocation is shown in Appendix A.

### 4.2. Irritable Bowel Syndrome (IBS) Model

Mice were anesthetized using 1% Rompun (100 μL, intraperitoneally (i.p.); Bayer, Seoul, Republic of Korea) and 2% Zoletil (100 μL, i.p.; Virbac S.A., Carros, France). A plastic catheter [outer diameter = 4 mm] was inserted into the descending colon of the mouse to a depth of 4–6 cm from the anus, and TNBS (P2297, Sigma-Aldrich, 50 μL of 5% (*w*/*v*) TNBS solution diluted using 50% ethanol, i.r.) was slowly instilled. To ensure uniform TNBS distribution within the entire colon, mice were carefully held in a vertical position for 1 min after injection. Control mice received 100 μL of 50% ethanol solution using the same procedure. The experimental setup is shown in Figure 1A. The colon tissue and serum were collected four weeks after TNBS or 50% ethanol administration.

### 4.3. Atopic Dermatitis (AD) Model

AD was induced in mice four weeks after TNBS or 50% ethanol administration once PI/IBS was established. Briefly, mice were anesthetized with ethyl ether and the right cheek was shaved for at least two days before treatment with MC903 (Sigma-Aldrich, St Louis, MO, USA). To induce AD-like lesions, 20 μL of 100% ethanol alone or ethanol containing 0.55 μg MC903 was applied to the shaved skin once a day for 7 days. AD severity tests were performed 13 days after the first MC903 application (6 days after the last application). Mice were anesthetized with ether and were then sacrificed to minimize animal suffering. The skin, distal colon, and serum samples were collected after behavioral tests.

### 4.4. Modified SCORAD Score

AD severity was assessed [39] using the modified SCORAD score on Day 13. The severity of each symptom was scored as 0 (absence), 1 (mild), 2 (moderate), and 3 (severe). This scoring was based on the severity of (1) erythema/hemorrhage, (2) scaling/dryness, and (3) edema. The sum of the individual scores (minimum 0; maximum 9) was used as the dermatitis score. Skin assessment was performed by an investigator who was blinded to the animal groups.

### 4.5. Scratching Behavior

The scratching behavior test was performed [40] on Days 5 (ongoing treatment) and 10 (post treatment). The time spent scratching the application site was quantified over the 30 min period. One bout of scratching was defined as an episode in which a mouse lifted its hind paw and scratched continuously for any length of time until the paw returned to the floor. Blind evaluation was performed by two skilled researchers having no information on the mice groups, and the results were verified by comparing them.

### 4.6. Stool Collection

On Day 12 of MC903 treatment, stools were collected by placing the mice in covered translucent cylinders (12 cm diameter × 25 cm high) for 1 h. The stool pellets were collected, weighed (T1), and left for drying at room temperature for 3 days. The dried pellets were weight again (T2). Stool water content was measured using the following formula: (T1 − T2)/total output pellet.

### 4.7. Bead Expulsion Test

On Day 13 of MC903 treatment, the mice were subjected to overnight fasting after which the animals were provided with free access to water. The colonic transit time in all mice groups was measured using the bead expulsion test. A 3 mm glass bead was inserted into the colon [2 cm proximal to the anus] using a plastic Pasteur pipette lightly lubricated with lubricating jelly, as previously described [41]. The time until bead expulsion was recorded.

### 4.8. Hematoxylin and Eosin Staining

Tissue samples were fixed in 10% formalin and embedded in paraffin. Paraffin-embedded skin and colon samples were sectioned (8 μm thick) and stained with H&E for histological analysis. Sections were imaged using a microscope (BX53; Olympus Corporation, Tokyo, Japan). The degree of epidermal hyperplasia was evaluated by measuring the epidermal thickness using ImageJ software (1.53k, National Institutes of Health, Bethesda, MD, USA).

### 4.9. Quantitative RT-PCR

Total RNA from the skin and colon was isolated using the TRIzol reagent (Invitrogen). The isolated RNA (2 μg) was incubated for 1 h at 42 °C using the SuperScript III First-Strand kit (Invitrogen, Waltham, MA, USA) and amplified using the KAPA Taq PCR kit (Kapabiosystems, Wilmington, MA, USA) according to manufacturer’s instructions. All primer sequences are presented in Appendix A. Real-time PCR was performed using the LightCycler^®^ Nano Instrument (Roche Molecular Systems, Inc., Branchburg, NJ, USA) with KAPA SYBR FAST qPCR Master Mix (Kapabiosystems, Wilmington, MA, USA), according to manufacturer’s instructions. β-actin was used for normalization and as an internal control.

### 4.10. In Vivo Intestinal Permeability Test

Intestinal permeability was measured [42] using dextran-FITC. The water bottles were taken out of the cage the day before. The following day, each mouse (10 mg/25 g body weight) was injected orally with FITC dextran which was dissolved in phosphate-buffered saline at a dosage of 50 mg/mL. Four hours after dosing, mice were anesthetized with isoflurane, and blood was collected via the left ventricle. After centrifugation, plasma was obtained, and the concentration of FITC dextran was measured using a GloMax^®^ Discover Microplate Reader (Promega, Madison, WI, USA) with excitation at 485 nm and emission at 535 nm.

### 4.11. Evaluation of Inflammatory Cytokine Levels

Four weeks following TNBS administration, 10 days following fecal microbiota transplantation (FMT), or 13 days following MC903 application, blood samples were collected from all mice and centrifuged at 12,000 rpm for 15 min. The supernatants were collected and stored at −70 °C. Serum was analyzed using the ELISA OptEIA kits [BD Bioscience, San Jose, CA, USA] for IL-6 and TNF-α according to the manufacturer’s instructions. 

### 4.12. Endotoxin Assay

LPS concentration in the serum was evaluated using the ToxinSensor™ Chromogenic LAL Endotoxin Assay kit (GenScript Inc., Piscataway, NJ, USA) according to the manufacturer’s instructions.

### 4.13. Immunofluorescence Analysis

Immunofluorescence staining was performed [43] to evaluate ZO-1 and occludin expression in the colon. Anti-TSLP (rabbit, 1:50; GTX85059, GeneTex, Irvine, CA, USA), anti-IL-6R (rabbit, 1:1000; Invitrogen, Waltham, MA, USA), anti-ZO-1 (rabbit, 1:500; 61–7300, Invitrogen, Waltham, MA, USA), and anti-occludin (mouse, 1:200; OC3F10, Invitrogen, Waltham, MA, USA) antibodies were diluted in 1× PBST supplemented with 0.3% BSA. The slides were wrapped in an aluminum foil to block light and stored at 4 °C for 72 h. Tissue sections were incubated for 1 h with a mixture of Alexa 488-conjugated donkey anti-mouse secondary antibody (1:1000; A21206, Invitrogen, Waltham, MA, USA) or Alexa 594-conjugated donkey anti-mouse secondary antibody (1:1000; A21203, Invitrogen, Waltham, MA, USA).

### 4.14. 16S rRNA Gene Sequencing

Total DNA was extracted from one fecal pellet using the PowerMax Soil DNA Isolation Kit [MO BIO Laboratories, Carlsbad, CA, USA]. The V3-V4 regions of the 16S rRNA gene were sequenced using the Illumina MiSeq platform (Macrogen, Republic of Korea). The sequencing data were processed using the Qiime2 (V 2021.8) pipeline with DADA2 as the denoising method [44]. Additionally, the EzTaxon database was used to annotate taxonomic information. An average of 24,286 reads were obtained per sample, Shannon and inverse Simpson diversity indices were chosen as parameters for alpha diversity. Beta diversity was measured using unweighted UniFrac and weighted UniFrac phylogenetic distance matrices, as well as Bray–Curtis. The Mann–Whitney U test was applied when comparing variables of two categories, and the Kruskal–Wallis test was used for multiple group comparisons. Significant differences in the relative abundance of bacteria at different taxonomic levels between groups were detected using linear discriminant analysis (LDA) effect size (LEfSe). Only LDA values > 2.5 at *p*-value < 0.05 were considered significantly enriched. The LEfSe results were visualized using heat maps, taxonomic bar charts, and cladograms.

### 4.15. The IBS Patient and Healthy Control

Experiments on human-derived materials were approved by the Kyung Hee University Hospital at Gangdong (KHNMCOH2021-11-006-008). Upon enrollment, written consent was obtained from each patient. Consent from patients or guardians was not necessary since the inclusion criteria stated eligible participants to be 20 years or older. The patients with moderate to severe IBS-D [diarrhea-predominant] had to be diagnosed according to the Rome IV criteria (recurrent abdominal pain or discomfort, or an uncomfortable sensation not described as pain at least three days a month in the past three months, associated with two or more of the following: improvement with defecation, onset associated with a change in frequency of stool, and onset associated with a change in the form appearance of stool. The criteria should be fulfilled for the past three months with symptoms onset at least six months before diagnosis). The patients diagnosed with IBS-D had >25% loose/wet motions (Bristol stool scale 6–7). IBS severity was determined using the IBS Severity Scoring System (IBS-SSS). Healthy control (HC) was defined as a person with a score of 50 or less on the IBS-SSS (Appendix A).

The following alarm features were required to be absent during screening to minimize the risk of missing important organic diseases: rectal bleeding, anemia, unexplained weight loss, nocturnal diarrhea, and a family history of organic GI diseases (e.g., colon cancer or inflammatory bowel disease). The patient agreed to not start any other treatment unless clinically indicated. Exclusion criteria were treatment with probiotics within the last three months, concurrent severe illness (cancer, uncontrolled diabetes mellitus, hepatic, renal, or cardiac dysfunction, and hyper- or hypothyroidism), previous GI surgery, chronic organic bowel disorders (e.g., inflammatory bowel disease, tuberculosis, diverticular disease, etc.), treatment with antibiotics during the two months prior to enrolment, pregnancy, or lactation.

### 4.16. Fecal Microbiota Transplantation

Recipient mice were treated by gavage with antibiotics (300 µL; 1 mg/mL, gentamycin; 1 mg/mL ampicillin; 1 mg/mL, neomycin; 1 mg/mL, metronidazole; 0.5 mg/mL, vancomycin) for 10 days prior to injection of the bacterial suspension. Fresh fecal pellets were collected from mice 4 weeks after TNBS administration as well as from patients with IBS and HC. The feces were pooled, soaked in sterile PBS (one fecal pellet/mL; human feces 2 g/20 mL) for approximately 15 min, homogenized, and centrifuged at 1000 rpm for 5 min at 4 °C to pellet the particulate matter. The suspension was again centrifuged at 5000 rpm for 5 min at 4 °C to obtain total bacteria. The final bacterial suspension was mixed with sterile glycerol (final concentration 20%) and stored at −80 °C until transplantation [45]. The suspensions were diluted to an OD at 600 nm of 0.5 in sterile PBS, corresponding to approximately 10^8^ CFU/mL [46]. Recipient mice were administered with 200 µL of bacterial suspension using a gavage for 10 days.

### 4.17. Statistical Analysis

All statistical analyses were performed using the GraphPad Prism software (version 9.4.0; San Diego, CA, USA). Scratching behavior, modified SCORAD score, epidermal thickness, stool weight, water content, bead expulsion latency, immunohistochemistry, Western blotting, and ELISA data were subjected to unpaired two-tailed *t*-test or one-way ANOVA followed by Newman–Keuls post hoc test. Body weight was analyzed using two-way repeated-measures ANOVA followed by Bonferroni’s post hoc test. All data are expressed as mean ± standard error of the mean (SEM). In all analyses, *p* < 0.05 was considered statistically significant.

## Figures and Tables

**Figure 1 ijms-25-02753-f001:**
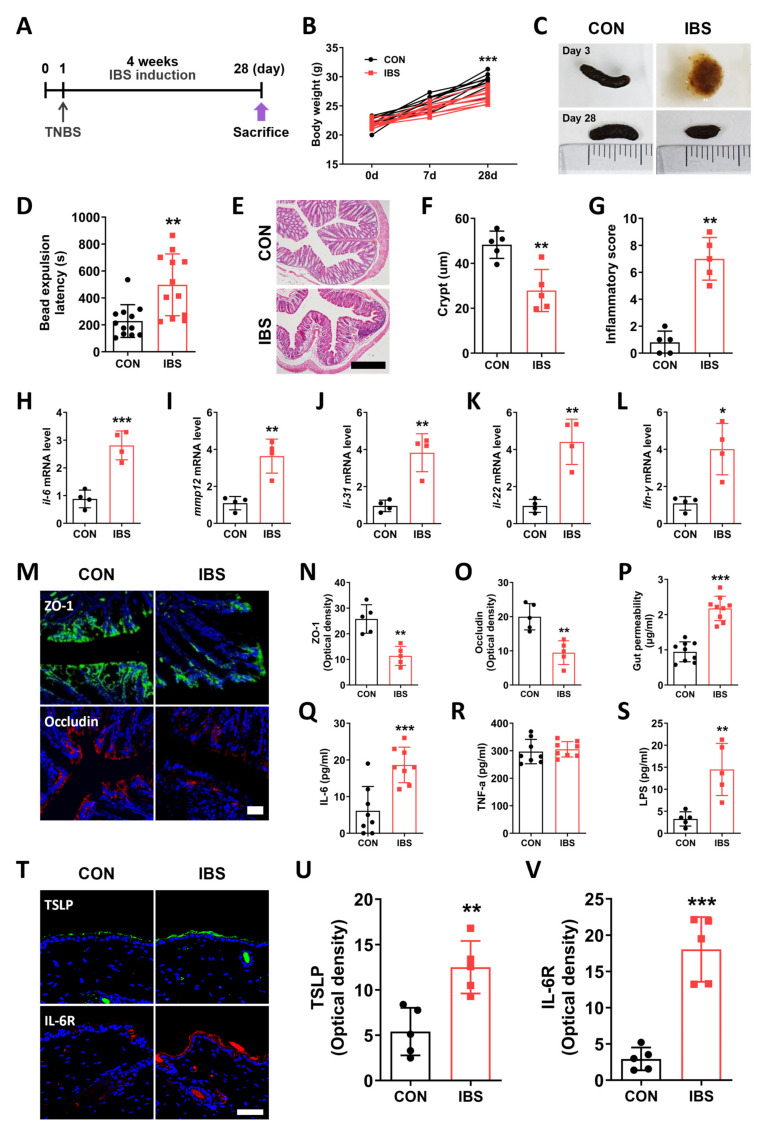
Changes in the gut–blood–skin axis in 2,4,6-trinitrobenzenesulfonic acid (TNBS)-induced IBS mouse. (**A**) Schematic diagram showing the experimental setup. Gastrointestinal functions were evaluated by analyzing body weight (**B**), stool morphology (**C**), and bead expulsion test (**D**). (**E**–**G**) Length of crypts and inflammatory score in CON and IBS groups (scale bar: 100 μm). mRNA levels of *Il-6* (**H**), *Mmp12* (**I**), *Il-31* (**J**), *Il-22* (**K**), *Ifn-r* (**L**) in the colon. (**M**–**O**) Expression levels of tight junction proteins ZO-1 and occludin in the colon of CON and IBS group mice (scale bar: 30 μm). Gut permeability was measured using the FITC-dextran test (**P**). Serum levels of IL-6 (**Q**), TNF-α (**R**), and LPS (**S**) in all mice groups. (**T**–**V**) TSLP and IL-6R expression in the mice skin from different groups (scale bar: 30 μm). *n* = 5–8/group. * *p* < 0.05, ** *p* < 0.01, *** *p* < 0.001 vs. CON. Data were analyzed by two-way ANOVA followed by Bonferroni’s post hoc test or unpaired two-tailed *t*-test.

**Figure 2 ijms-25-02753-f002:**
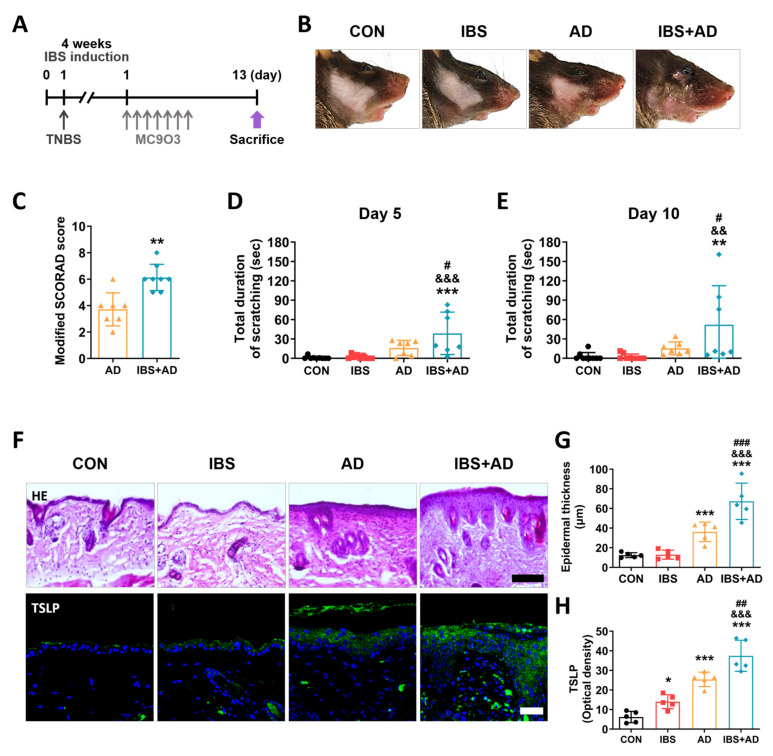
Atopic dermatitis (AD) symptoms are exacerbated in IBS mice. (**A**) Schematic diagram showing experimental design. (**B**,**C**) Skin lesions and modified SCORAD score on Day 13. Results of the scratching behavior test on Days 5 (**D**) and 10 (**E**). (**F**–**H**) Epidermal thickness (scale bar: 100 μm) and TSLP expression (scale bar: 30 μm) in the mice skin from different groups. *n* = 5–8/group. * *p* < 0.05, ** *p* < 0.01, *** *p* < 0.001 vs. CON, ^&&^
*p* < 0.01, ^&&&^
*p* < 0.001 vs. IBS, ^#^ *p* < 0.05, ^##^ *p* < 0.01, ^###^ *p* < 0.001 vs. AD. All data were analyzed by one-way ANOVA followed by Newman–Keuls test or unpaired two-tailed *t*-test.

**Figure 3 ijms-25-02753-f003:**
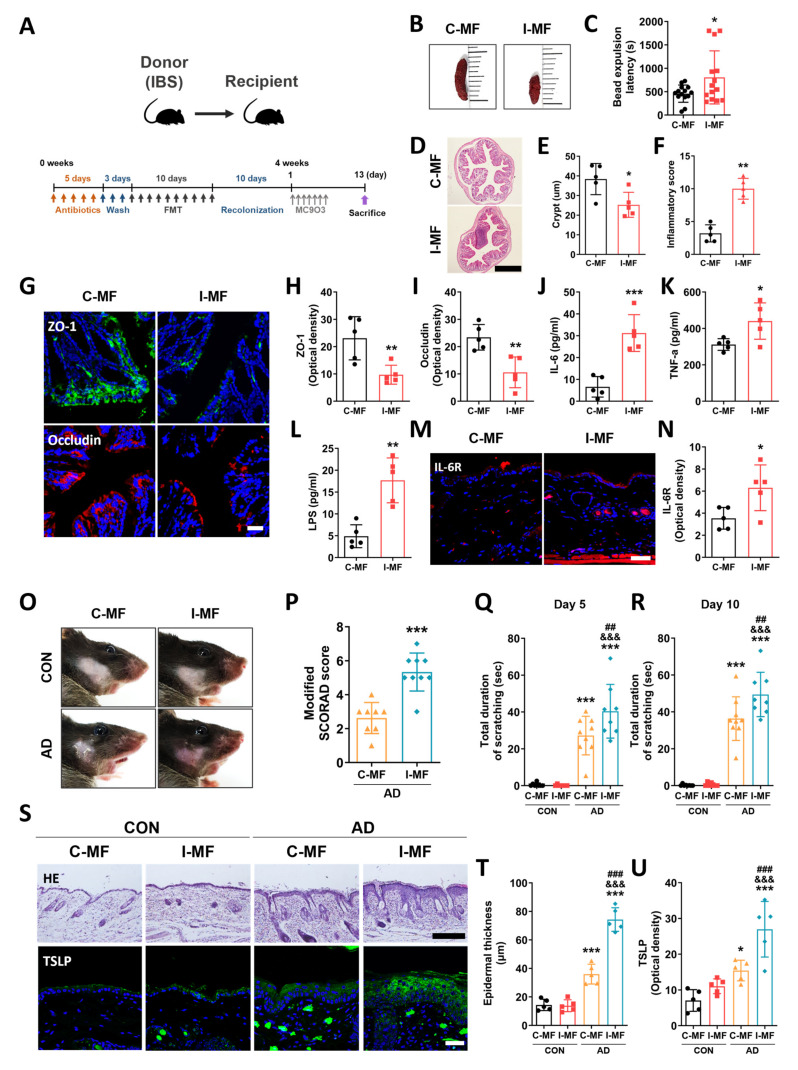
AD symptoms are exacerbated in healthy mice transplanted with gut microbiota from IBS mice. (**A**) Schematic diagram showing the experimental design. Gastrointestinal functions were evaluated using the stool morphology examination (**B**) and the bead expulsion test (**C**). (**D**–**F**) Length of crypts (scale bar: 200 μm), inflammatory score, and (**G**–**I**) expression levels of ZO-1 and occludin in the colon of mice from different groups (scale bar: 30 μm). Serum levels of IL-6 (**J**), TNF-α (**K**), and LPS (**L**) in different mice groups. (**M**,**N**) Expression levels of IL-6R in the mice skin (scale bar: 30 μm). * *p* < 0.05, ** *p* < 0.01, *** *p* < 0.001 vs. I-MF. (**O**,**P**) Skin lesions and modified SCORAD score on Day 13. Results of the scratch-ing behavior test on Days 5 (**Q**) and 10 (**R**). (**S**–**U**) Epidermal thickness (scale bar: 100 μm) and expression levels of TSLP in the mice skin (scale bar: 30 μm) from different groups. *n* = 5–14/group. * *p* < 0.05, *** *p* < 0.001 vs. CON+C-MF, ^&&&^
*p* < 0.001 vs. CON+I-MF, ^##^
*p* < 0.01, ^###^
*p* < 0.001 vs. AD+C-MF. All data were analyzed by one-way ANOVA followed by Newman–Keuls test or unpaired two-tailed *t*-test.

**Figure 4 ijms-25-02753-f004:**
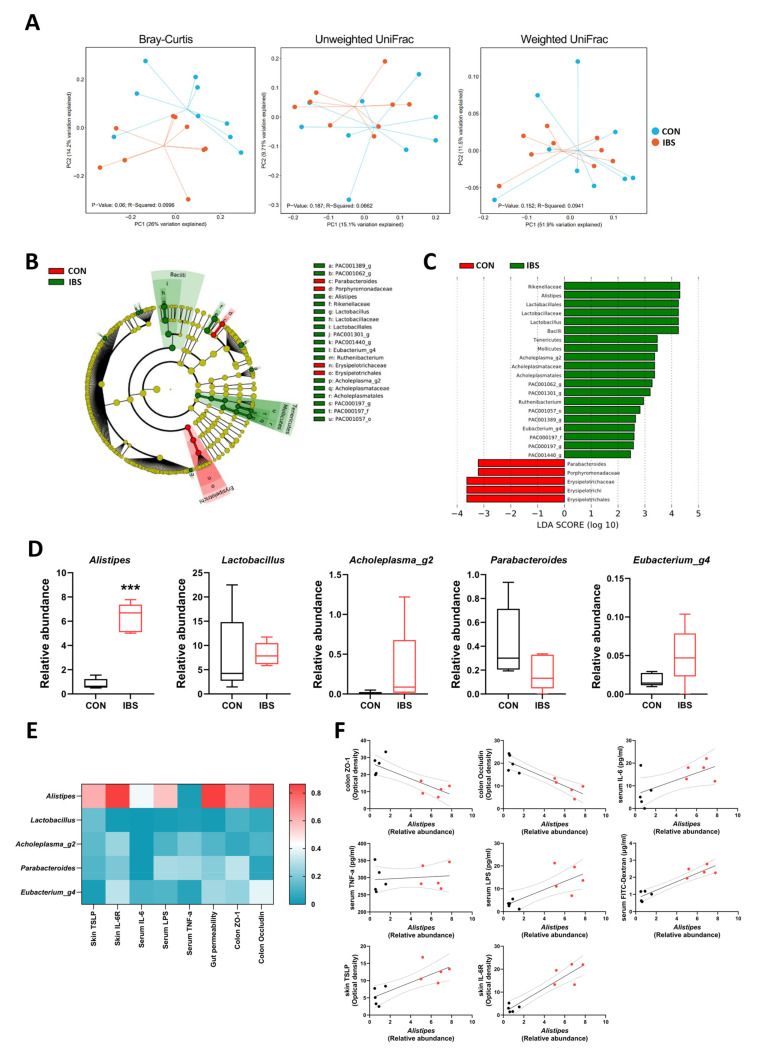
Gut microbial dysbiosis and its association with gut–blood–skin axis markers in IBS mice. (**A**) Two−dimensional principal component analysis (PCoA) plots of Bray–Curtis, unweighted UniFrac, and weighted UniFrac distances generated using the relative abundance of OTUs. Each point represents a sample from CON and IBS groups. Comparison of the gut microbiota using LES between CON and IBS groups at the OTU level. Cladograms (**B**) and the histogram (**C**) of bacterial taxa that differed significantly between the two groups (LDA > 3.0, *p* < 0.05). The red, green, and blue shading in the clado-gram depicts bacterial taxa that were significantly higher in either the CON group or in the IBS group, as indicated. c, class; o, order; f, family; g, genus; s, species. *n* = 9 per group. The relative abundance of 5 bacterial genera (**D**). (**E**) Heat map shows the correlation among gut−blood−skin axis markers at the genus level. (**F**) Correlations between relative abundance of *Alistipes* and the gut-blood-skin markers. Black circle: CON group; Red circle: IBS group. *n* = 5/group. *** *p* < 0.001 vs. CON. All data were analyzed by unpaired two-tailed *t*-test.

**Figure 5 ijms-25-02753-f005:**
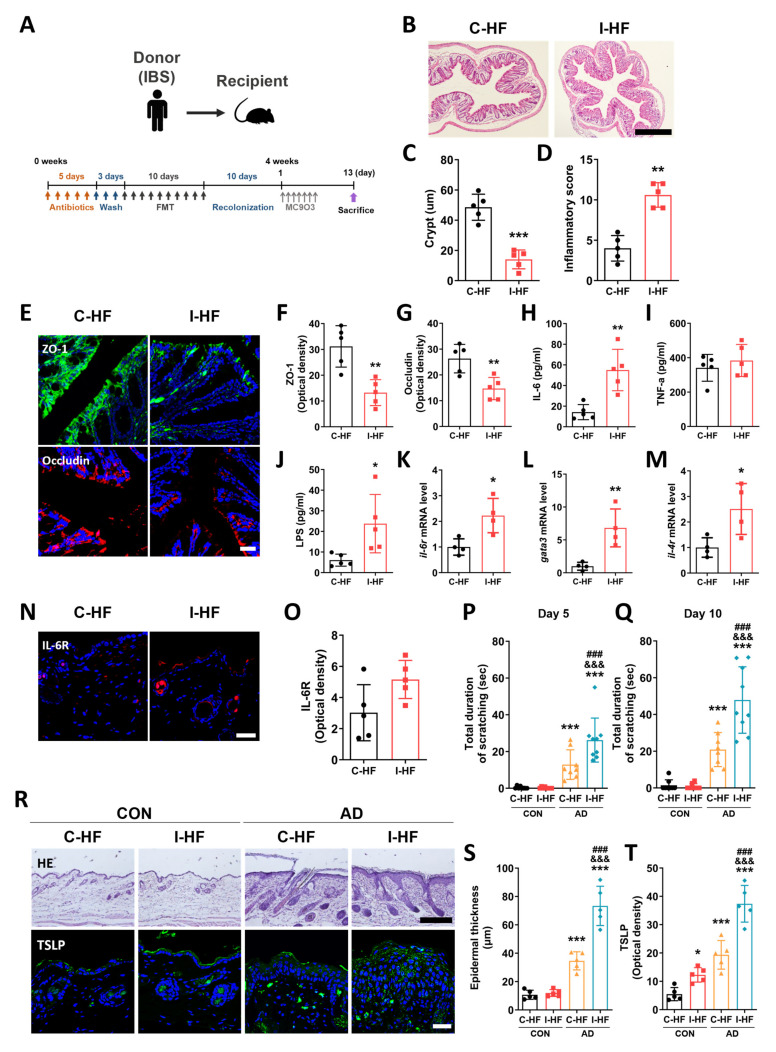
AD symptoms are exacerbated in the mice transplanted with gut microbiota from the IBS patient. (**A**) Schematic diagram showing the experimental setup. (**B**–**D**) Length of crypts, inflammatory score (scale bar: 200 μm), and (**E**–**G**) ZO-1 and occludin levels in the colon of C-HF and I-HF group mice (scale bar: 30 μm). Serum levels of IL-6 (**H**), TNF-α (**I**), and LPS (**J**) in different mice groups. Expression levels of il-6r (**K**), gata3 (**L**), and il-4r (**M**) transcripts in the skin. (**N**,**O**) Expression of IL-6R in the skin of different mice groups (scale bar: 30 μm). * *p* < 0.05, ** *p* < 0.01, *** *p* < 0.001 vs. C-MF. Results of the scratching behavior test on Days 5 (**P**) and 10 (**Q**). (**R**–**T**) Epidermal thickness (scale bar: 100 μm) and expression levels of TSLP (scale bar: 30 μm) in the skin. *n* = 4–9/group. * *p* < 0.05, *** *p* < 0.001 vs. CON+C-HF, ^&&&^ *p* < 0.001 vs. CON+I-HF, ^###^ *p* < 0.001 vs. AD+C-HF. All data were analyzed by one-way ANOVA followed by Newman–Keuls test or unpaired two-tailed *t*-test.

## Data Availability

The data presented in this study are available on request from the corresponding author.

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
