# Peer review of "Chronic Gut Inflammation and Dysbiosis in IBS: Unraveling Their Contribution to Atopic Dermatitis Progression"

_ijms, 2024, doi:10.3390/ijms25052753_

Round 1

Reviewer 1 Report

Comments and Suggestions for Authors

Review of paper Chronic Gut Inflammation and Dysbiosis in IBS: Unravelling Their Contribution to Atopic Dermatitis Progression”.

75-96 is not specified how the groups are composed, how many mice are used and the modality/duration of the trial.

216 → “ANO-VA” there is “-“ Please remove

249 → “Com-ponent” there is “-“ Please remove

252 → “mi-crobiota” there is “-“ Please remove

336 → “to investigated” it is an error with the English, it need to be changed in <To investigate>

489 → (#555240) and (#555268) = I don’t understand at what this code refer to (if they are part of the test or other)

559 → “transplantation21” = maybe the number need to be written like this <(21)>

574 → The titles of the supporting information are missing

Comments on the Quality of English Language

Minor editing of English language required.

Overall english is okay.

Author Response

[Q1] 75-96 → is not specified how the groups are composed, how many mice are used and the modality/duration of the trial.

[Answer]

Thank you for your comments. Based on your suggestions, we have incorporated detailed information about the groups into the main text (lines 88-91). Furthermore, for more comprehensive details, we have included 'S2 Tables of Detailed information on group allocation' as follows:

Supplementary Table S2. Detailed information on group allocation

Experiment 1: Establishment of 2,4,6-trinitrobenzenesulfonic acid (TNBS)-induced IBS mouse.

Group

Treatment

CON

Ethanol-injected group

IBS

TNBS-injected group

Experiment 2: Changes of AD symptoms in TNBS-induced IBS mouse

Group

Treatment

Injection into the colon

On the right cheek

CON

50% ethanol

100% ethanol

IBS

TNBS

100% ethanol

AD

50% ethanol

MC903

IBS+AD

TNBS

MC9O3

Experiment 3: Changes of AD symptoms in mice transplanted with gut microbiota from IBS mice

Group

Treatment

FMT

On the right cheek

CON

C-MF

transplanted with fecal microbiota from the CON mice

100% ethanol

I-MF

transplanted with fecal microbiota from the IBS mice

AD

C-MF

transplanted with fecal microbiota from the CON mice

MC903

I-MF

transplanted with fecal microbiota from the IBS mice

Experiment 4: Changes of AD symptoms in mice transplanted with gut microbiota from the IBS patient.

Group

Treatment

FMT

On the right cheek

CON

C-HF

transplanted from the gut microbiota of healthy control

100% ethanol

I-HF

transplanted from the gut microbiota of the IBS patient

AD

C-HF

transplanted from the gut microbiota of healthy control

MC903

I-HF

transplanted from the gut microbiota of the IBS patient

[Q2]

  1. 216 → “ANO-VA” there is “-“ Please remove
  2. 249 → “Com-ponent” there is “-“ Please remove
  3. 252 → “mi-crobiota” there is “-“ Please remove
  4. 336 → “to investigated” it is an error with the English, it need to be changed in <To investigate>
  5. 489 → (#555240) and (#555268) = I don’t understand at what this code refer to (if they are part of the test or other)
  6. 574 → The titles of the supporting information are missing.

[Answers]

Thank you for your meticulous attention to detail in reviewing our manuscript. We have carefully reviewed the points you highlighted and have made the following revisions:

  1. Removed the hyphen in “ANO-VA” to correct it to “ANOVA” (line 229).
  2. Removed the hyphen in “Com-ponent” to correct it to “Component” (line 262).
  3. Removed the hyphen in “mi-crobiota” to correct it to “microbiota” (line 265).
  4. Corrected the phrase “to investigated” to “to investigate” for proper grammatical structure (line 349).
  5. Deleted the product numbers (line 510).
  6. Added the titles of the supporting information as follows:

Table S1: Primer for real-time PCR; Table S2: Detailed information on group allocation; Table S3: Demographics and characteristics of donor (HC and IBS).

[Q3]

559 → “transplantation21” = maybe the number need to be written like this <(21)>

[Answer]

Thank you for your comments. Due to a program error, the reference was not properly inserted. We have verified and added the correct references as follows:

The final bacterial suspension was mixed with sterile glycerol (final concentration 20%) and stored at -80 °C until transplantation [45]. The suspensions were diluted to an OD at 600 nm of 0.5 in sterile PBS, corresponding to approximately 108 CFU/mL [46].

[45] Hamilton, M.J.; Weingarden, A.R.; Sadowsky, M.J.; Khoruts, A. Standardized frozen preparation for transplantation of fecal microbiota for recurrent Clostridium difficile infection. Am J Gastroenterol 2012, 107, 761-767, doi:10.1038/ajg.2011.482.

[46] Schuijt, T.J.; Lankelma, J.M.; Scicluna, B.P.; de Sousa e Melo, F.; Roelofs, J.J.; de Boer, J.D.; Hoogendijk, A.J.; de Beer, R.; de Vos, A.; Belzer, C.; et al. The gut microbiota plays a protective role in the host defence against pneumococcal pneumonia. Gut 2016, 65, 575-583, doi:10.1136/gutjnl-2015-309728.

Reviewer 2 Report

Comments and Suggestions for Authors

Thank you for allowing me to review this paper. In the study, the authors investigate the changes in pro-inflammatory cytokines and histological alterations in the gut and skin of an IBS mouse model. The results are striking, clearly demonstrating a correlation between the exacerbation of atopic dermatitis, IBS, and the microbiota.

The study is well-conducted, methodologically sound, with very proficient use of English, and I did not observe any significant methodological issues. The methods are comprehensively detailed and the narrative is well-structured.

From my perspective, here are the suggested corrections:

1) I recommend expanding the discussion to include findings from existing literature on IBS, atopic dermatitis, and the microbiota. There are additional data available that could provide a broader context. Comparing the authors' data with these pre-existing studies would be beneficial.

2) In the discussion section, a paragraph should be devoted to addressing the potential and existing data from similar studies found in the literature. The study has its limitations and should consider the differences in application to human subjects.

3) It's well-known that the microbiota influences not only intestinal diseases but also other illnesses. I suggest adding a note about the role of the microbiota in the introduction and also discussing its role in other extra-intestinal diseases in the discussion section. Are the pathological mechanisms the same across these diseases? What about the potential therapies? 

Comments on the Quality of English Language

NA

Author Response

[Q1]

I recommend expanding the discussion to include findings from existing literature on IBS, atopic dermatitis, and the microbiota. There are additional data available that could provide a broader context. Comparing the authors' data with these pre-existing studies would be beneficial.

[Answer]

Thank you for your comments. Following your suggestion, we have included findings from existing literature on IBS, atopic dermatitis, and the microbiota as follows:

In the introduction, Lines 64-71:

A notable example is irritable bowel syndrome (IBS), characterized by persistent gut inflammation and changes in gut microbiota [7]. Research indicates that in conditions such as IBS, gut inflammation can lead to increased intestinal permeability. This change allows microbial products and pro-inflammatory cytokines to enter the systemic circulation, possibly contributing to widespread inflammation [8]. This mechanism could exacerbate conditions like atopic dermatitis (AD). The higher prevalence of AD in patients with gut inflammatory diseases, including IBS, underscores the importance of the gut-skin axis [9].

[Q2]

In the discussion section, a paragraph should be devoted to addressing the potential and existing data from similar studies found in the literature. The study has its limitations and should consider the differences in application to human subjects.

[Answer]

We appreciate your comments. Considering the challenges of extrapolating our findings to human subjects, we have added limitations to our study as follows:

In the discussion, Lines 399-409:

While our study provides substantial evidence linking gut inflammation and microbial dysbiosis with AD development, several limitations remain. The precise mechanisms of how gut dysbiosis contributes to systemic inflammation and the detailed role of the gut-skin axis in AD pathogenesis need further exploration. Additionally, understanding the interaction between inflammatory factors like IL-6 and IL-6R in AD's development is crucial. In addition, the extrapolation of results from an IBS mouse model to human AD patients requires cautious interpretation. While our findings offer valuable insights, the complexities of human physiology and the multifactorial nature of AD mean that these results should be considered a starting point for further research. Additionally, the exact mechanisms by which gut microbial dysbiosis and inflammation influence skin conditions in humans remain to be fully elucidated.

[Q3] It's well-known that the microbiota influences not only intestinal diseases but also other illnesses. I suggest adding a note about the role of the microbiota in the introduction and also discussing its role in other extra-intestinal diseases in the discussion section. Are the pathological mechanisms the same across these diseases? What about the potential therapies?

[Answer]

Thank you for your comments. Based on your suggestion, we have incorporated a note regarding the role of the gut microbiota in the introduction.

In addition, we commented that the changes in gut microbiota were shown in abdominal pain in other studies: Alistipes, identified in the gut microbiota of IBS mice in our study, was also reported to be associated with abdominal pain. This may play a role in systemic inflammation and increased gut permeability, both of which are known to exacerbate AD features. We commented on this in the discussion. We also added the therapeutic potential of modulating gut microbiota in the discussion section.

In the introduction, Lines 61-64:

The gut microbiome, a complex ecosystem of microorganisms, is known to influence systemic immunity and inflammation beyond the gastrointestinal tract [6]. Gut microbial dysbiosis, an imbalance in this microbial community, has been linked to various inflammatory conditions, including those affecting skin diseases such as AD.

In the discussion, Lines 361-374:

Further analysis revealed a significant correlation between the altered gut microbiota, specifically Alistipes, and markers of gut integrity, gut permeability: its abundance was inversely correlated with the integrity of tight junctions in the gut epithelia and positively correlated with increased gut permeability, as well as elevated serum levels of LPS and IL-6. In addition, we observed that the altered gut microbiota can be the prognostic factors to predict the pathogenesis of AD: the abundance of Alistipes was positively associated with the expressions of TSLP in the skin, suggesting a close influence of gut microbiota on skin inflammation. This aligns with previous literature indicating that Alistipes is associated with abdominal pain [28] and may play a role in systemic inflammation and increased gut permeability [29-31], both of which are known to exacerbate AD features. These finding is particularly intriguing as it links a specific microbial genus with physiological changes relevant to exacerbation of AD. Nevertheless, further in-depth research is essential to elucidate the specific role that Alistipes plays in the aggravation of atopic dermatitis.

In the discussion, Lines 379-382:

Overall, our study presents novel insights by demonstrating that gut inflammation and gut dysbiosis shown in IBS can exacerbate AD, while previous studies have explored their association [11,12,32]. Thus, we suggest that modulating gut microbiota could be a new therapeutic approach for AD.

Reviewer 3 Report

Comments and Suggestions for Authors

Dear authors, 

I think that your study  is very interesting and very inportant considering high incidence of AD. Gur microbiote was analyzed  for many disorders. 

I have some suggestions:

you can inprove introduction section with major details of AD;

You can insert limitations section;

You can improve the effective novelty of your study to AD.

Author Response

[Q1] You can improve introduction section with major details of AD

[Answer]

Thank you for your comments. Following your suggestion, we have incorporated additional details about AD into the introduction section as follows:

In the introduction, Lines 45-59:

Atopic dermatitis (AD), a chronic inflammatory skin disease, has been increasingly associated with systemic health conditions, particularly gastrointestinal disorders [1]. It is primarily characterized by skin erythema, edema, and dryness, driven by an increased T-helper (Th) 2 cell-mediated immune response [1]. The etiology of AD involves a complex interplay of genetic, environmental, epidermal barrier dysfunction, and abnormal immune responses [2]. Furthermore, recent epidemiological studies have elucidated the prevalence and impact of AD globally, emphasizing its rising incidence and the burden it places on individuals and healthcare systems [3].

Emerging research indicates that the etiology of AD, while multifactorial, is strongly influenced by epidermal barrier dysfunction and abnormal immune responses. Intriguingly, emerging research underscores the significant role of gut health, particularly the gut microbiota balance, in the pathogenesis and exacerbation of AD [4]. Changes in gut microbiota associated with AD involve a reduction in beneficial bacteria and an increase in pathogenic microbes, potentially contributing to systemic inflammation and skin barrier dysfunction in AD patients.

[Q2] You can insert limitations section

[Answer]

Thank you for your comments. In the discussion section, we have provided a description of the limitations of our study as follows:

In the discussion, Lines 399-409:

While our study provides substantial evidence linking gut inflammation and microbial dysbiosis with AD development, several limitations remain. The precise mechanisms of how gut dysbiosis contributes to systemic inflammation and the detailed role of the gut-skin axis in AD pathogenesis need further exploration. Additionally, understanding the interaction between inflammatory factors like IL-6 and IL-6R in AD's development is crucial. In addition, the extrapolation of results from an IBS mouse model to human AD patients requires cautious interpretation. While our findings offer valuable insights, the complexities of human physiology and the multifactorial nature of AD mean that these results should be considered a starting point for further research. Additionally, the exact mechanisms by which gut microbial dysbiosis and inflammation influence skin conditions in humans remain to be fully elucidated.

[Q3] You can improve the effective novelty of your study to AD.

[Answer]

Thank you for your comments. We have explicitly presented the novelty of our study, highlighting its significance as follows:

In the discussion, Lines, 379-382:

Overall, our study presents novel insights by demonstrating that gut inflammation and gut dysbiosis shown in IBS can exacerbate AD, while previous studies have explored their association [11,12,32]. Thus, we suggest that modulating gut microbiota could be a new therapeutic approach for AD.

In the discussion, Lines, 410-413:

Our research highlights the intricate relationship between chronic gut inflammation, microbial dysbiosis, and the exacerbation of AD. These findings provide a foundation for future investigations into the gut-skin axis and open up new possibilities for treating AD by targeting gut health.